# Discovery of 1,2,4-Oxadiazole Derivatives Containing Haloalkyl as Potential Acetylcholine Receptor Nematicides

**DOI:** 10.3390/ijms24065773

**Published:** 2023-03-17

**Authors:** Ling Luo, Yuqin Ou, Qi Zhang, Xiuhai Gan

**Affiliations:** National Key Laboratory of Green Pesticide, Key Laboratory of Green Pesticide and Agricultural Bioengineering, Ministry of Education, Guizhou University, Guiyang 550025, China

**Keywords:** 1,2,4-oxadiazole derivatives, tioxazafen, nematocidal activity, transcriptome, acetylcholinesterase

## Abstract

Plant-parasitic nematodes pose a serious threat to crops and cause substantial financial losses due to control difficulties. Tioxazafen (3-phenyl-5-thiophen-2-yl-1,2,4-oxadiazole) is a novel broad-spectrum nematicide developed by the Monsanto Company, which shows good prevention effects on many kinds of nematodes. To discover compounds with high nematocidal activities, 48 derivatives of 1,2,4-oxadiazole were obtained by introducing haloalkyl at the 5-position of tioxazafen, and their nematocidal activities were systematically evaluated. The bioassays revealed that most of 1,2,4-oxadiazole derivatives showed remarkable nematocidal activities against *Bursaphelenchus xylophilus*, *Aphelenchoides besseyi*, and *Ditylenchus dipsaci*. Notably, compound **A1** showed excellent nematocidal activity against *B. xylophilus* with LC_50_ values of 2.4 μg/mL, which was superior to that of avermectin (335.5 μg/mL), tioxazafen (>300 μg/mL), and fosthiazate (436.9 μg/mL). The transcriptome and enzyme activity results indicate that the nematocidal activity of compound A1 was mainly related to the compound which affected the acetylcholine receptor of *B. xylophilus*.

## 1. Introduction

Plant-parasitic nematodes (PPNs) are difficult to control because of their small size, many species, wide range of hosts, and fast transmission, which can cause economic losses of up to USD 157 billion annually worldwide [1,2,3]. As three important PPNs, *Bursaphelenchus xylophilus*, *Aphelenchoides besseyi*, and *Ditylenchus destructor* can severely harm the healthy growth of pine trees [4], rice, and potato yields [5,6], respectively. Chemical nematicides play a crucial role in nematode management strategies because of their rapid response to PPNs [7,8]. However, most current commercial nematicides are mainly highly toxic organophosphorus and carbamate, such as cadusafos (2-(butan-2-ylsulfanyl-ethoxy-phosphoryl), sulfanylbutane (2-butan-2-ylsulfanylbutane), fenamiphos (ethyl 3-methyl-4-(methylthio)phenyl (1-methylethyl)phosphoramidate ethyl 3-methyl-4-(methylthio)phenyl (1-methylethyl)phosphoramidate), fosthiazate ((2-Oxo-3-thiazolidinyl)phosphonothioic acid *O*-ethyl *S*-(1-methylpropyl) ester), dazomet(3,5-dimethyl-1,3,5-thiadiazinane-2-thione), oxamyl (*O*-methylaminoformyl-1-dimethylcarbamoyl-1-methylthioaldoxime), and aldicarbl (Amino formyl *O*-methyl-2-methyl-2-(methyl sulfonium) propionic aldehyde oxime, 2-methyl-2-(methyl sulfonium) propionic aldehyde-*O*-[(methyl amino) carbonyl] oxime). Due to resistance development and environmental protection requirements, some of these have been withdrawn from the market, including methyl bromide, 1,2-dibromo-3-chloropropane, and aldicarb [9,10,11]. As a successful nematicide, avermectin has been widely used to control nematodes in the field. However, the control effect is not satisfactory because of its instability [12]. Therefore, the development of high-efficiency and low-risk nematicides is still the key scientific problem of nematode control [13,14,15].

It is well known that halogen atoms or halogen-containing substituents not only possess unique steric effects [16], electronic effects [17], bond polarization, and pKa-value [18], but also improve the metabolic, oxidative, thermal stability [19,20], and lipophilicity [21]. These physicochemical properties can improve the activity of drugs. Thus, the introduction of halogen atoms into pesticides is a reasonable way to develop environmentally friendly, highly active, quick, and economically feasible agricultural products in agricultural chemistry [22]. At present, abundant pesticides on the market contain halogen atoms [23]. In particular, most of the registered nematicides contain halogen atoms, such as fluensulfone, trifluorocide, acetoprole, fluazaindolizine, fluopyram, and cylcobutrifluram (Figure 1), which also contain individual chlorine atoms, fluorine atoms, trifluoromethyl, or “mixed” halogens.

Five-membered heterocyclics are widely used in molecular design for developing new drugs [24]. As an important heterocyclic, 1,2,4-oxadiazole exhibits a wide range of biological activities, including antifungal [25,26], herbicidal [27], anti-inflammatory [28,29], and insecticidal activities [30,31]. Tioxazafen is a new broad-spectrum nematicide containing a 1,2,4-oxadiazole heterocycle designed by Monsanto Company for controlling systemic nematodes in soybean, corn, and cotton [32]. It can interfere the ribosome of nematodes and eventually cause the death of nematodes [33,34]. However, tioxazafen lacks flexibility in its molecular structure and is currently only used as a seed treatment agent. In our previous works, a number of 1,2,4-oxadiazole derivatives were designed and synthesized based on tioxazafen, but failed to find the lead compound with good nematocidal activity [25,35]. In order to continue the derivation of tioxazafen, we introduced an active fragment with a haloalkyl group at the 5-position of 1,2,4-oxadiazole (Figure 2). The in vitro bioassays indicated that compound **A1** (5-(chloromethyl)-3-(4-fluorophenyl)-1,2,4-oxadiazole) exhibited excellent nematocidal activity against *B. xylophilus*, which was significantly higher than that of commercial avermectin, fosthiazate, and tioxazafen. In addition, the mode of action results demonstrated that compound **A1** affected the acetylcholine receptor of *B. xylophilus*.

## 2. Result and Discussion

### 2.1. Chemistry

As shown in Figure 3, forty-eight 1,2,4-oxadiazole derivatives were synthesized from aromatic cyanogen. First, the intermediates **1** were obtained following the known procedure with modification, in a yield exceeding 80% [36]. Then, the target compounds **A1**–**A29** and **B1**–**B19** were obtained with intermediates **1** and acyl chloride or acyl bromide. The structures of all compounds were identified by ^1^H NMR, ^13^C NMR, and HRMS. The physical properties and spectrogram data of these derivatives are provided in Appendix A.

### 2.2. Nematocidal Activity

As shown in Table 1, most of the compounds demonstrated high nematocidal activities against *B. xylophilus*, *A*. *besseyi*, and *D*. *dipsaci*. Among them, compounds **A1**–**A15**, **A17**, **A18**, and **B1**–**B13** exhibited remarkable nematocidal activities against *B. xylophilus* at 50 μg/mL for 48 h, with a corrected mortality of 100%. In order to further evaluate the nematocidal activity of the compounds against *B. xylophilus*, the concentration was reduced to 10 μg/mL. As a result, the nematocidal activity of most compounds was significantly reduced. However, compounds **A1**, **A2**, **A3**, **A4**, **B1**, **B2**, **B3**, **B4**, **B5**, **B6**, **B7**, and **B8** showed significant nematocidal activities against *B. xylophilus*, with LC_50_ values of 2.4, 2.8, 3.3, 4.2, 2.6, 3.5, 3.5, 3.7, 3.8, 4.5, 4.8, and 4.9 μg/mL, respectively, stronger than that of fosthiazate (436.9 μg/mL), avermectin (335.5 μg/mL), and tioxazafen (>300 μg/mL), but less than that of fluopyram (0.9 μg/mL). Notably, compound **A1** displayed the best nematocidal activity against *B. xylophilus*. The mortality of compound **A1** at 10 μg/mL for 48 h is shown in Figure 4. Meanwhile, compounds **A1**–**A15**, **B1**–**B11**, and **B16** showed significant nematocidal activity against *A. besseyi*, with a corrected mortality of 100% at 50 μg/mL for 48 h. Furthermore, compounds **A6**, **A7**, **B2**, **B4**, **B5**, **B8**, **B10**, **B11**, and **B16** showed 100% mortality against *A. besseyi* at 10 μg/mL. The LC_50_ values of some compounds were tested, as displayed in Table 2. The results reveal that compound **A6** displayed the best nematocidal activity against *A. besseyi*, with the LC_50_ value of 3.8 μg/mL, superior to that of fosthiazate (388.5 μg/mL), avermectin (56.8 μg/mL), and tioxazafen (142.9 μg/mL), and comparable to that of fluopyram (1.5 μg/mL). In addition, compounds **A1**–**A7**, **A9**–**A19**, **B1**–**B11**, and **B13** demonstrated remarkable nematocidal activity against *D. dipsaci*, with a corrected mortality of 100% at 50 μg/mL for 48 h. Among these compounds, compound **A7** exhibited outstanding nematocidal activity against *D. dipsaci*, with an LC_50_ value of 2.7 μg/mL, which was better than that of fosthiazate (333.3 μg/mL), avermectin (285.4 μg/mL), and tioxazafen (>300 μg/mL).

The results of the structure–activity relationship analysis showed that the introduction of chloromethyl or bromomethyl in the 5-position of 1,2,4-oxadiazole enhanced the nematocidal activity of the compound. Among these, the compounds with the substituted 4-position on the benzene ring possessed excellent nematocidal activity, such as **B1** (Ar = 4-CH_3_-C_6_H_4_, X = Cl, LC_50_ = 2.6 μg/mL) > **B15** (Ar = 2-CH_3_-C_6_H_4_, X = Cl, LC_50_ = 31.9 μg/mL), and **A4** (Ar = 4-Cl-C_6_H_4_, X = Br, LC_50_ = 4.2 μg/mL) > **A16** (Ar = 2-Cl-C_6_H_4_, X = Br, LC_50_ = 13.4 μg/mL). When the benzene ring was substituted with F, diF, and triF atoms, respectively, the nematocidal activity against *B. xylophilus* of the compound **A1** (Ar = 4-F-C_6_H_4_, X = Cl, LC_50_ = 2.4 μg/mL) > **A12** (Ar = 2,4-diF-C_6_H_3_, X = Cl, LC_50_ = 8.2 μg/mL) > **A26** (Ar = 2,4,5-triCl-C_6_H_2_, X = Cl, LC_50_ = 133.4 μg/mL), indicating that steric hindrance had a certain effect on nematocidal activity. Meanwhile, if the number of halogens was increased or the carbon chain was prolonged, the activity was greatly reduced, for example, the nematocidal activity against *B. xylophilus* of compound **B1** (Ar = 4-CH_3_-C_6_H_4_, X = Cl, LC_50_ = 2.6 μg/mL) > **B16** (Ar = 4-CH_3_-C_6_H_4_, X = diCl, LC_50_ = 60.6 μg/mL), and **A2** (Ar = 4-Cl-C_6_H_4_, X = Cl, LC_50_ = 2.8 μg/mL) > **A25** (R = 4-Cl-C_6_H_4_, X = CH_2_Cl, LC_50_ = 89.9 μg/mL). When the 5-position of the 1,2,4-oxadiazole derivative was replaced by chloromethyl, the nematocidal activity of the 1,2,4-oxadiazole derivative against *B. xylophilus* was better than that of the bromomethyl-substituted compound, which was also evidenced by the following activity order as **A3** (Ar = 4-Br-C_6_H_4_, X = Cl, LC_50_ = 3.3 μg/mL) > **A11** (Ar = 4-Br-C_6_H_4_, X = Br, LC_50_ = 7.0 μg/mL). Furthermore, when the 4-position of the benzene ring was the electron-withdrawing substituent, the nematocidal activity of the compound against *B. xylophilus* nematodes was gradually weakened with the increase in the atomic radius, which was supported by the following order as **A1** (Ar = 4-F-C_6_H_4_, X = Cl, LC_50_ = 2.4 μg/mL) > **A2** (Ar = 4-Cl-C_6_H_4_, X = Cl, LC_50_ = 2.8 μg/mL) > **A3** (Ar = 4-Br-C_6_H_4_, X = Cl, LC_50_ = 3.3 μg/mL), which explains why compound **A1** had excellent nematocidal activity. In conclusion, in order to improve the nematocidal activity, the 5-position of 1,2,4-oxadiazole must contain chloromethyl or bromomethyl.

### 2.3. DEGs Analysis

Through sequencing analysis, a total of 58,292 DEGs were identified in the **A1** and CK (control check, the test solution that did not contain compound) treatments. Compared to CK treatment, there were 7395 upregulated genes and 9918 downregulated genes of the **A1** treatment (Appendix A). The DEGs were subjected to the functional annotation of the database. A total of 17,313 DEGs were marked, with 3976 and 16,380 annotated in the GO and KEGG functional databases, respectively (Appendix A). For the sake of the GO functional classification, most of the downregulated and upregulated DEGs were mainly enriched in biological processes (BPs) and molecular functions (MFs) rather than in cellular components (CCs) (Figure 5). The downregulated and upregulated DEGs were mainly concentrated in cellular, metabolic, single-organism, and localization processes in the BP ontology but mainly distributed in cell, cell part, membrane, and organelle in the MF ontology. In the CC categories, catalytic activity and binding were mostly enriched. These upregulated and downregulated DEGs were related to cellular processes, cellular metabolism, multicellular development, cellular senescence, oxidative stress response, and lipid metabolism, suggesting that **A1** affected the metabolism, development, and aging of *B. xylophilus*.

In addition, the KEGG pathway analysis indicated that 299 pathways were annotated. The top 20 pathways were demonstrated (*p* < 0.05, Figure 6), including cholinergic synapse (ko04725), metabolism of xenobiotics by cytochrome P450 (ko00980), drug metabolism—cytochrome P450 (ko00982) encoding different hydrolases, drug metabolism—other enzymes (ko00983), and so forth (Appendix A). The results showed that compound **A1** obviously affects the cholinergic synapse pathway (Figure 7). The expression of key genes including muscarinic acetylcholine receptor (*mAChR*), choline acetyltransferase (*ChAT*), vesicular acetylcholine transporter (*vACht*), and acetylcholin esterase (*AChE*) showed to be downregulated. Among these, the downregulated expression of *mAChR* can produce a parasympathetic nerve excitation effect [37]. Meanwhile, *ChAT* can transfer acetyl coenzyme A to choline and then form acetylcholine [38], *vACht* was the rate-limiting factor of acetylcholine transport in the presynaptic membrane and one of the main specific markers of the cholinergic system [39]. As a key enzyme in nerve conduction, AChE can hydrolyze acetylcholin to stop the excitatory effect of this neurotransmitter on the postsynaptic membrane and ensure the normal transmission of nerve signals in vivo [40], and it can be seen as an important target for nematicides [41]. The downregulated expression of *ChAT*, *vACht*, and *AChE* genes indicated that the synthesis of these enzymes was inhibited. Therefore, compound **A1** may downregulate the expression levels of *ChAT*, *vACht*, and *AChE* genes, and the cause abnormal nerve signal transmission of *B. xylophilus*.

### 2.4. qRT-PCR Validation

Differential expressions of three DEGs related to cholinergic synapses including *AChE1*, *AChE2*, and *ChAT* were measured. As shown in Figure 8, the expression level of *AChE1* in **A1** treatment improved by 1.04 times compared to the CK treatment. However, the expression levels of *AChE2* and *ChAT* in **A1** treatment were both downregulated. Especially, the expression level of *ChAT* decreased 11.5 times compared to the CK treatment. Differential expressions of three DEGs measured by qRT-PCR were highly consistent with those shown by RNA-seq. It indicated that the transcriptome-based differentially expressed genes analysis was highly reliable.

### 2.5. Enzymatic Activity of AChE

AChE is a crucial enzyme that guarantees the normal operation of the nervous system in the organism and is affected by nAChR [42,43]. Transcriptomic analysis showed that compound **A1** may bind to the nAChR receptor. In order to confirm the potential mechanism of our derivatives, we measured the activity of **A1** to the AChE enzyme. The effect of compound **A1** on AChE content was determined at different times and concentrations (Figure 9).

It can be seen from the figure that compound **A1** exhibited a strong inhibitory effect on AChE and that the intensity of inhibition was basically positively correlated with time and concentration. With the increase in the concentration or the extension of time, the AChE content gradually decreased. In particular, at 15 μg/mL for 48 h, the content of AChE was the lowest, i.e., 0.8 U mg^−1^. However, at 15 μg/mL and at 10 μg/mL for 6 h, the content of AChE temporarily increased. It was speculated that the concentration was so high that the odor emitted by compound **A1** stimulated *B. xylophilus* to produce a stress response, thus resulting in the temporary increase in the target enzyme. AChE can rapidly hydrolyze the neurotransmitter acetylcholine (ACh), thus terminating the transmission of nerve impulses [44]. Once acetylcholinesterase was inhibited, the nerve impulse of the nematode cannot be stopped, resulting in the nematode being excited to death. These results indicate that compound **A1** may act on the nervous system of *B. xylophilus*, which was consistent with the results of KEGG pathway analysis.

### 2.6. Molecular Docking and MD Simulations

#### 2.6.1. Molecular Docking

To explain the binding mode of compound **A1** to the AChE, a molecular docking study on compound **A1** was conducted, and tioxazafen was selected as the comparative standard. As seen in Figure 10A,B, compound **A1** and tioxazafen can be well connected to the surrounding amino acid residues of the active pocket through Pi–Pi stacked interactions, and they demonstrate similar conformations in the active protein pocket on the AChE. The benzene ring and 1,2,4-oxadiazole ring of two compounds can form Pi–Pi stacked interactions with residues TRP-84 and PHE-330. In addition, the fluorine and chlorine atoms of **A1** can form stronger and weaker hydrogen bonds with residues GLY-118 and TYR-334, respectively, which were critical to the stability of the combination of AChE inhibitors and AChE. However, tioxazafen has not observed this interaction.

#### 2.6.2. MD Simulations

In MD simulations, it was found that compound **A1** and fosthiazate formed complexes to AChE proteins. Figure 11 shows that the RMSD values of compound **A1** and fosthiazate were similar after 3 ns and fluctuated between 0.4 and 0.6 nm, which indicated that the system reached equilibrium and that the ligand was stably bound to this active site of the protein. However, when the RMSD of the ligand alone was monitored, **A1** was found to be slightly higher than fosthiazate, despite its relative stability after 2 ns. In order to explain this, we combined the free energy calculations (Table 3). According to Table 3, the reason was the larger −*T*ΔS of fosthiazate owing to the fact that its structure had more movable bonds, which enhanced the structural flexibility, while **A1** did not, which was also the reason for which its RMSD did not change significantly after self-equilibration. Moreover, the ΔG_bind_ of **A1** was higher than that of fosthiazate, which was consistent with our activity test results. However, **A1** had a greater hydrophobic interaction with the protein in molecular docking, which was also reflected in a larger ΔE_vdw_ than fosthiazate because of the solid aromatic ring of **A1**, thus leading to a lower **A1** than fosthiazate in ΔE_ele_ and ΔG_sol_ because of fewer hydrogen bond acceptors and hydrogen bond donors for **A1.**

## 3. Materials and Methods

### 3.1. Instruments and Chemicals

The reaction process was monitored by thin-layer chromatography (TLC) with a 254 nm UV detector. Column chromatography was performed on 200–300 mesh silica gel. The melting points were recorded using an uncorrected X-4B microscope melting point apparatus (Shanghai Electrophysical Optical Instrument Co., Ltd., Shanghai, China). ^1^H NMR and ^13^C NMR detections were performed at room temperature with CDCl_3_ as a solvent on the Bruker DPX-400 or DPX-500 spectrometers (Bruker, Billerica, MA, USA). Tetramethylsilane was used as an internal standard, and high-resolution mass spectrometer (HRMS) data for the compound were tested in positive mode on the Thermo Scientific Q Exactive (Thermo Scientific, St. Louis, MO, USA) mass spectrometer. All chemical reagents, aromatic nitrile, heterocyclic nitrile, and other raw material compounds were purchased from the supplier and used directly without further purification, unless otherwise stated.

#### 3.1.1. General Procedure for the Synthesis of Intermediate 1

A 50% aqueous solution of sodium hydroxide (1.5 mmol) was added to an ethanolic solution of hydroxylamine hydrochloride (1.5 mmol) and stirred at room temperature for 10 min. Differently substituted benzonitrile (1.0 mmol) or heterocyclic nitrile was dissolved in ethanol, added to the above mixture, and then warmed to 80 °C and stirred for 7 h. The reaction was monitored by means of thin-layer chromatography (TLC). After the reaction was completed, the generated solid was removed by filtration, the filtrate was concentrated under vacuum, and then the mixture was washed with saturated NaCl solution (30 mL), extracted with ethyl acetate (30 mL) three times, dried with anhydrous Na_2_SO_4_, filtered, and concentrated under vacuum to give the intermediate **1** in yields of above 80.0–95.0%.

#### 3.1.2. General Procedure for the Synthesis Compounds **A1**–**A29** and **B1**–**B19**

Acyl chloride or acyl bromide (1.5 mmol) with different substituents were added to the toluene solution of intermediate **1** (1.0 mmol) and triethylamine (1.5 mmol) in an ice bath. The reaction mixture was stirred continuously in an ice bath for 10 min and then transferred to reflux at 110 °C until intermediate **1** was consumed. After the reaction was completed, the organic layer was extracted with saturated NaCl solution and ethyl acetate. After being dried with anhydrous Na_2_SO_4_, the solvent was removed in vacuum. The target compounds **A1**–**A29** and **B1**–**B19** were then purified by column chromatography.

### 3.2. Nematocidal Activity Assay

*B*. *xylophilus*, *D*. *dipsaci*, and *A*. *besseyi* were cultivated on potato dextrose agar–Botrytis cinerea provided by the Fine Chemical Research and Development Center of Guizhou University (Guiyang, China). Nematocidal bioassays were systematically carried out on all compounds using conventional methods with slight modifications [45,46]. A total of 1 mg of the target compound was weighed and added with 100 μL DMF for dissolution (the final concentration of DMF was 0.5%). The pipette was used to measure 20 and 4 μL, and then 1% Tween-80 was fixed to 4 mL to obtain concentrations of 50 and 10 μg/mL, respectively. Subsequently, 10 μL of nematode suspension containing 80 nematodes and 0.3 mL of a test solution were added to a 48-well biochemical culture dish for testing. Each treatment was repeated three times. The commercially available nematicides fosthiazate, avermectin, tioxazafen, and fluopyram were used as positive controls at 50 and 10 μg/mL, and the test solution that did not contain the compound was the negative control. The entire experiment was repeated three times. Nematodes treated under a microscope for 48 h were considered dead if they did not move and straighten. The corrected mortality was calculated with the follow formula. LC_50_ was tested in the same way.
Corrected mortality % = [(mortality of treatment % − mortality of negative control %)/(1 − mortality of negative control %)] × 100

### 3.3. Transcriptome Profiling

*B. xylophilus* was treated with **A1** at 5 μg/mL for 48 h. Meanwhile, the solution that did not contain the compound was used as a CK. The treated nematodes were collected and stored in a −80 °C freezer. Total RNA of **A1** and *B. xylophilus* that was not treated with compound was extracted with Trizol reagent. RNA purity and integrity were detected by 1% agarose gel electrophoresis, and high-quality RNA was used for transcriptome sequencing. High-quality mRNA was enriched using Oligo (dT) magnetic beads. Subsequently, the mRNA fragment was used as a template to synthesize the first-strand cDNA using random hexamer primers and reverse transcriptase (RNAse H), and then the second-strand cDNA was synthesized using DNA polymerase I, RNAse H, buffer, and dNTPs. Double-stranded cDNA was purified using AMPUREXP beads, end-repaired, a-tailed, and connected to the sequencing adapter. AMPure XP beads were used to select the fragment size and finally enriched by PCR to obtain a cDNA library [47]. The number of gene fragments per kilobase per million bases was calculated based on the length of the gene and the number of reads mapped to the gene. DEGs were scored using a threshold of Padj < 0.05 and |log_2_ fold change| > 1 [48]. Finally, the enrichment of DEGs was analyzed by the Gene Ontology (GO) and Kyoto Encyclopedia of Genes and Genomes (KEGG) pathway analysis method.

### 3.4. Quantitative Real-Time Polymerase Chain Reaction

Total RNA was extracted from plant tissue samples using RNApure plant kit (Cwbio, Beijing, China) and then used to synthesize cDNA using the PrimeScript^TM^RT reagent kit (TaKaRa, Beijing, China). Quantitative real-time polymerase chain reaction (qRT-PCR) was performed using the TB Green Premix Ex Tap^TM^ Ⅱ kit (TaKaRa, Beijing, China), targeting *AChE1*, *AChE2*, and *AChT* genes. *Actin* was also included as an internal amplification control. The reaction was performed using a CFX96 system (Bio-Rad, Hercules, CA, USA) under the conditions described by the reagent’s instructions. The primer sequences for target genes are outlined in Table 4. All mRNA levels were normalized by the *Actin* level [49]. Gene expression was relatively quantitated by the 2^−ΔΔCT^ method.

### 3.5. Enzyme Activity of AChE

In order to explore the possible target of compound **A1**, a series of concentration gradients and time gradients were set to detect the activity of acetylcholinesterase (AChE). The activity of AChE was calculated using an enzyme-linked immunosorbent assay kit (Beijing Solarbio Science & Technology Co., Ltd., Beijing, China) according to the manufacturer’s instructions. After treating *B. xylophilus* with different concentrations of compound **A1**, approximately 100 mg weight of nematodes was collected at 6, 12, 24, and 48 h, respectively. After that, nematodes were quickly fixed in liquid nitrogen, thoroughly ground in mortar, and then added with corresponding extracts and fully homogenized in an ice bath. The mixture was centrifuged (8000 rpm/min, 10 min) and the supernatant was obtained at 4 °C. The activity of AChE in supernatant was detected. The enzyme activity value was calculated by (U/g) = 3758 × ΔA/W (ΔA = A Measurement tube − A control tube, W = quality).

### 3.6. Molecular Docking and MD Simulations

Preliminary experiments showed that compound **A1** had a certain inhibitory effect on the AChE. To investigate the binding mode of **A1** and AChE, we carried out the molecular docking of **A1** and AChE. However, the AChE of *B. xylophilus* lacks a three-dimensional structure. Tetronarce californica acetylcholinesterase (PDB ID: 6H14), obtained from the RCSB protein database (https://www.rcsb.org/, accessed on 23 December 2022) and further processed by adding hydrogen atoms and removing water molecules with PYMOL, was the most suitable template. To prepare and optimize the compound structures, the binding mode of the compounds to the AChE proteins was determined by the program LeDock using ChemOffice 2019. The conformations of fosthiazate and **A1** were then clustered using a root mean square deviation (RMSD) of 2 Å as the threshold value. These representative conformations were used as initial conformations for the further minimization and MD simulations. Using the Amber18 method, we applied the Amber94 and TIP3P force fields to proteins and water, the GAFF force field to small organic molecules, and the system was completed by adding sodium and chloride ions for electrical neutralization. Then, the system was minimized for 1000 steps using the steepest descent method and the next 2000 steps using the conjugate gradient method. Then, the system was simulated by molecular dynamics according to the following steps: The system was heated up from 20 to 300 K under isovolumic conditions for 30 ps. Then, MD simulations were performed at 1 atm and 300 K by a relaxation process similar to the minimization. Finally, 10 ns molecular dynamics was performed for each system. The molecular mechanics Poisson–Boltzmann surface area (MM-PBSA) method was used to calculate the binding free energy (ΔGbind) of the sample compound to the AChE protein, and the contribution of residues to the ligand was determined using the deconstruction module. More details on three-step energy minimization and MM-PBSA can be found in the Appendix A.

### 3.7. Statistical Analysis

All data were analyzed by ANOVA using SPSS version 25 (IBM, Armonk, NY, USA) for statistical significance. The index and density data complied with normality according to the Kolmogorov–Smirnov test and variance homogeneity as determined with the Kruskal–Wallis test. All data were analyzed using paired *t*-test and Duncan’s multiple comparison test. The quantitative data were represented as means ± SE (standard error).

## 4. Conclusions

Forty-eight 1,2,4-oxadiazole derivatives containing haloalkyl were easily obtained and compound **A1** showed outstanding against *B. xylophilus*, with an LC_50_ value of 2.4 μg/mL, which was superior to those of commercial nematicides avermectin, tioxazafen, and fosthiazate. The transcriptomic and enzyme activity research results indicate that the nematocidal activity of compound **A1** was closely related to the acetylcholine receptor of *B. xylophilus*. It is worth mentioning that the synthesis method is simple and the target compounds are low in cost and could serve as cheap and potential acetylcholine receptor nematicides to control PPNs.

## Figures and Tables

**Figure 1 ijms-24-05773-f001:**
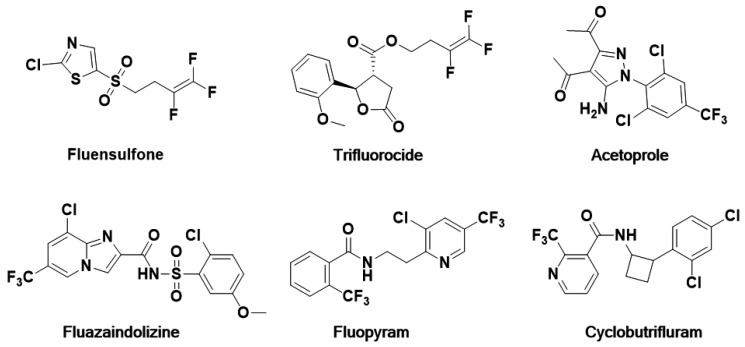
The structure of chemical nematicides containing halogen atoms.

**Figure 2 ijms-24-05773-f002:**
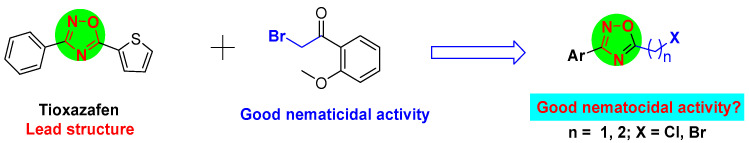
Design of the target compounds. The red font and green highlight represent the 1,2,4-oxadiazole potency. The blue part represents the haloalkyl groups introduced in favor of increased activity.

**Figure 3 ijms-24-05773-f003:**
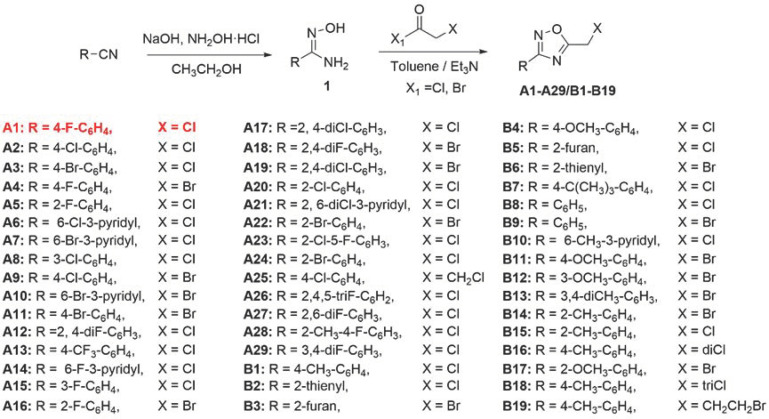
Synthetic procedure for 1,2,4-oxadiazole derivatives **A1–A29** and **B1–B19**. The red font represents the compound **A1** with the best activity against *B. xylophilus*.

**Figure 4 ijms-24-05773-f004:**
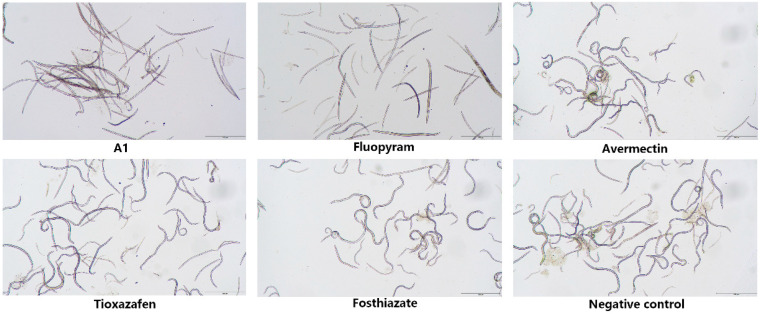
Nematocidal activity of compounds **A1** against *B. xylophilus* at 10 μg/mL for 48 h. All figures were imaged with a microscope at a scale of 500 m.

**Figure 5 ijms-24-05773-f005:**
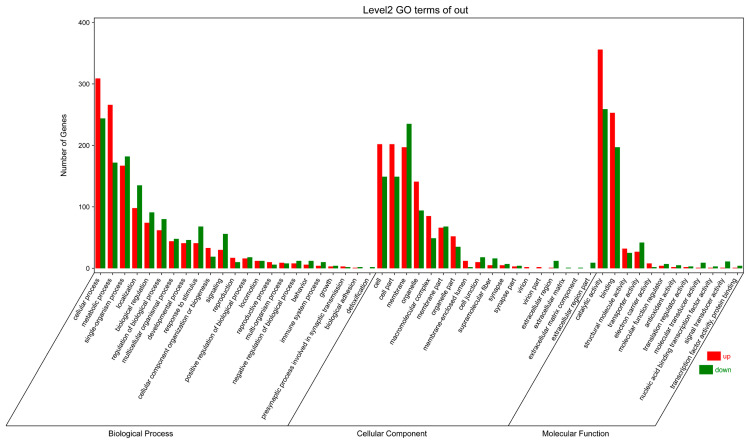
GO analysis of DEGs between **A1** treatment group and CK control group.

**Figure 6 ijms-24-05773-f006:**
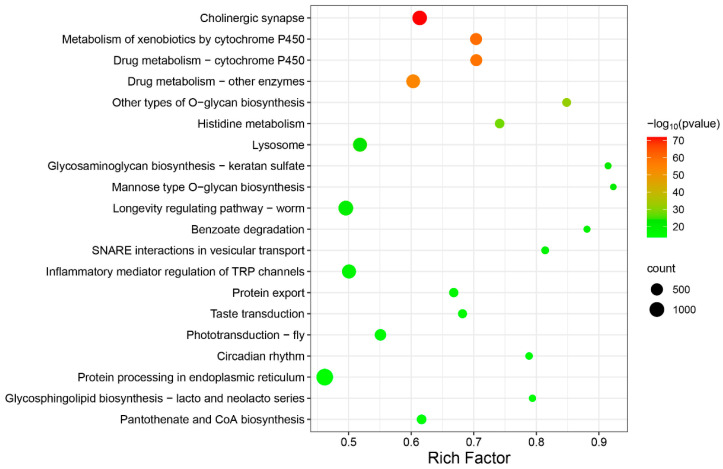
KEGG analysis of DEGs between **A1** treatment group and CK control group (*p* < 0.05, top 20).

**Figure 7 ijms-24-05773-f007:**
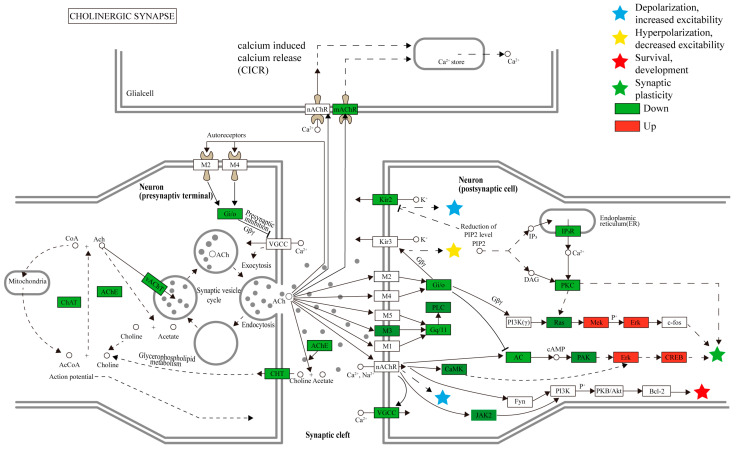
Cholinergic synapse pathway in *B. xylophilus* responded to **A1**. Up- and down-regulated genes are marked in red and green bars, respectively. Solid and dashed arrows represent direct and indirect positive regulation, respectively. Stripes indicate negative regulation.

**Figure 8 ijms-24-05773-f008:**
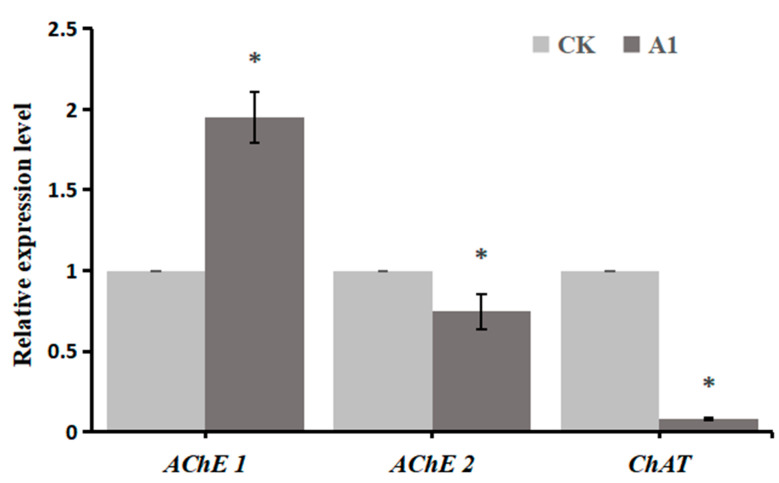
Expression analysis of genes related to the cholinergic synapses. Data were analyzed by ANOVA using SPSS. Data show the means ± standard deviation of three independent samples. * *p* < 0.05 vs. CK group.

**Figure 9 ijms-24-05773-f009:**
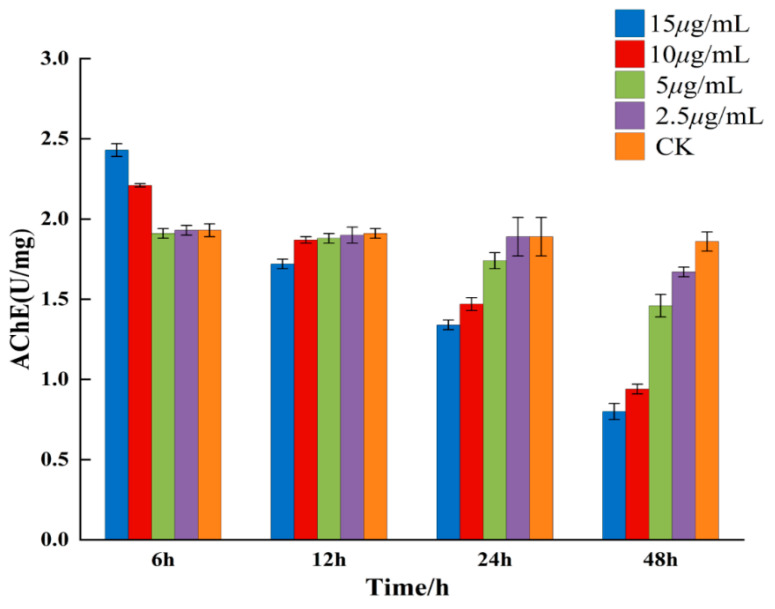
AChE activity of compound **A1** treatment groups and CK control group against *B. xylophilus*.

**Figure 10 ijms-24-05773-f010:**
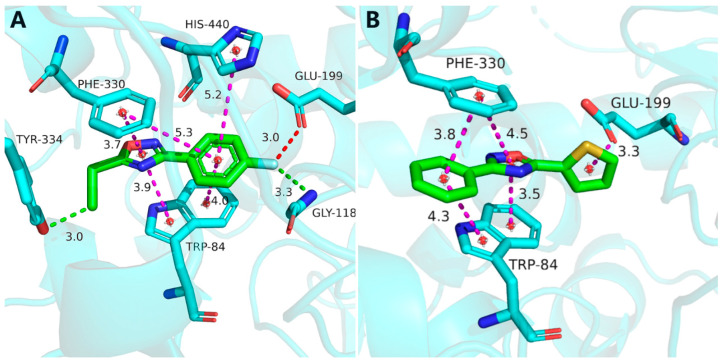
Molecular docking comparisons of the title compound **A1** (**A**) and tioxazafen (**B**) with AChE (PDB ID: ID: 6H14). The blue color of (**A**) represents nitrogen, blue-green represents carbon, green represents the structure of compound **A1**, red represents oxygen, in (**B**) blue represents nitrogen, blue-green represents carbon, green represents the structure of compound tioxazafen, yellow represents sulfur, red represents oxygen; The purple color of the dot line represents hydrophobicity, green represents hydrophilic action, and red represents halogen action.

**Figure 11 ijms-24-05773-f011:**
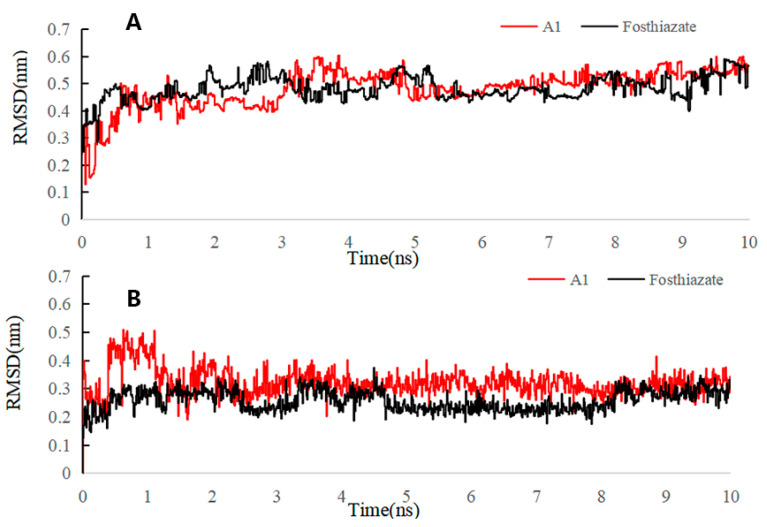
RMSD trajectories of AChE–ligand complex (**A**) and ligand (**B**) during 10 ns simulations.

**Table 1 ijms-24-05773-t001:** Nematicidal activity of compounds against *B. xylophilus*, *A. besseyi*, and *D. dipsaci*
^a^.

Compound	*B. xylophilus*			*A. besseyi*		*D. dipsaci*	
	50 μg/mL	10 μg/mL	LC_50_ μg/mL	50 μg/mL	10 μg/mL	50 μg/mL	10 μg/mL
**A1**	100	95.3 ± 2.8	2.4 ± 0.1	100	87.1 ± 4.8	100	91.2 ± 3.3
**A2**	100	90.0 ± 2.6	2.8 ± 0.1	100	60.5 ± 1.9	100	81.0 ± 1.6
**A3**	100	85.3 ± 5.9	3.3 ± 0.2	100	57.2 ± 9.1	100	80.4 ± 5.8
**A4**	100	100	4.2 ± 0.4	100	40.7 ± 4.6	100	92.8 ± 0.8
**A5**	100	63.2±8.1	5.0 ± 0.2	100	62.1 ± 7.1	100	80.5 ± 4.5
**A6**	100	73.4 ± 9.1	5.4 ± 0.2	100	100	100	97.4 ± 0.7
**A7**	100	100	5.5 ± 0.3	100	100	100	90.2 ± 3.0
**A8**	100	66.9 ± 8.6	6.5 ± 0.2	100	49.7 ± 9.8	94.8 ± 1.8	67.8 ± 4.2
**A9**	100	91.7 ± 9.0	6.6 ± 1.1	100	29.5 ± 8.3	100	90.5 ± 1.2
**A10**	100	75.0 ± 1.2	6.6 ± 1.2	100	23.9 ± 1.7	100	83.9 ± 3.5
**A11**	100	89.1 ± 6.2	7.0 ± 1.3	100	36.5 ± 2.4	100	87.9 ± 6.4
**A12**	100	67.0 ± 5.8	8.2 ± 0.3	100	47.7 ± 2.5	100	74.9 ± 4.2
**A13**	100	87.2 ± 1.0	8.6 ± 5.3	100	28.5 ± 2.0	100	70.6 ± 5.6
**A14**	100	58.7 ± 4.2	8.9 ± 1.3	100	65.0 ± 0.4	100	91.7 ± 4.3
**A15**	100	40.3 ± 1.7	11.2 ± 0.6	100	54.3 ± 7.5	100	67.1 ± 4.9
**A16**	87.0 ± 6.8	36.3 ± 3.9	13.4 ± 0.7	59.4 ± 1.0	17.1 ± 8.6	100	35.5 ± 7.9
**A17**	100	20.7 ± 3.3	24.9 ± 1.5	100	26.6 ± 3.8	100	26.2 ± 1.8
**A18**	100	21.8 ± 7.6	23.7 ± 2.5	90.6 ± 9.4	23.4 ± 3.0	100	43.8 ± 3.8
**A19**	80.6 ± 4.6	24.4 ± 4.3	26.6 ± 3.7	63.5 ± 9.0	11.8 ± 5.4	100	62.1 ± 7.6
**A20**	80.3 ± 4.4	27.5 ± 5.6	26.7 ± 0.5	70.5 ± 4.2	31.3 ± 9.2	77.4 ± 3.9	56.3 ± 1.9
**A21**	95.9 ± 0.7	36.1 ± 3.5	29.8 ± 0.9	32.0 ± 5.1	0	62.2 ± 4.1	0
**A22**	72.7±7.5	19.3 ± 3.1	36.1 ± 1.0	15.0 ± 6.0	0	78.1 ± 4.3	23.3 ± 8.8
**A23**	69.0 ± 8.4	0	39.7 ± 2.4	0	0	82.1 ± 1.5	18.8 ± 6.5
**A24**	45.2 ± 0.3	0	58.2 ± 6.9	0	0	61.5 ± 1.8	21.7 ± 9.7
**A25**	23.2 ± 6.3	0	89.9 ± 1.6	0	0	33.3 ± 5.1	0
**A26**	0	0	133.4 ± 5.9	0	0	25.2 ± 8.9	0
**A27**	0	0	156.7 ± 9.8	0	0	29.1 ± 6.0	16.9 ± 5.6
**A28**	0	0	203.3 ± 7.2	0	0	22.4 ± 8.6	0
**A29**	-	-	216.4 ± 7.9	-	-	43.8 ± 7.6	19.8 ± 2.8
**B1**	100	100	2.6 ± 0.7	100	89.9 ± 2.4	100	100
**B2**	100	100	3.5 ± 0.1	100	100	100	90.1 ± 5.5
**B3**	100	75.2 ± 8.1	3.5 ± 0.4	100	89.9 ± 2.0	100	86.3 ± 2.5
**B4**	100	78.2 ± 1.5	3.7 ± 0.3	100	100	100	86.9 ± 5.1
**B5**	100	100	3.8 ± 0.3	100	100	100	85.2 ± 7.0
**B6**	100	97.4 ± 4.4	4.5 ± 0.3	100	85.8 ± 3.2	100	87.0 ± 1.1
**B7**	100	80.1 ± 3.4	4.8 ± 0.2	100	72.0 ± 4.2	100	71.9 ± 3.8
**B8**	100	93.6 ± 3.9	4.9 ± 0.1	100	100	100	80.3 ± 3.6
**B9**	100	80.8 ± 4.6	6.9 ± 6.7	100	19.5 ± 1.5	100	88.2 ± 6.8
**B10**	100	58.3 ± 3.3	8.5 ± 0.5	100	100	100	66.7 ± 7.7
**B11**	100	75.5 ± 6.3	9.5 ± 0.8	100	30.7 ± 5.3	100	86.0 ± 5.8
**B12**	100	31.5 ± 0.8	13.3 ± 0.5	20.6 ± 6.6	0	94.6 ± 5.3	37.0 ± 9.6
**B13**	100	30.0 ± 4.4	14.3 ± 0.5	19.8 ± 9.7	7.3 ± 1.7	100	40.6 ± 9.1
**B14**	97.3 ± 2.5	0	15.2 ± 0.2	18.4 ± 4.3	0	68.2 ± 7.9	20.5 ± 2.4
**B15**	76.1 ± 5.0	0	31.9 ± 6.0	28.4 ± 7.7	0	67.2 ± 2.6	23.4 ± 4.7
**B16**	35.0 ± 3.6	0	60.6 ± 9.0	100	88.4 ± 2.3	51.3 ± 9.6	0
**B17**	38.9 ± 5.9	0	67.6 ± 5.8	30.5 ± 9.3	0	31.4 ± 3.6	22.7 ± 5.2
**B18**	0	0	158.5 ± 8.2	0	0	54.2 ± 3.3	23.4 ± 9.3
**B19**	0	0	200.0 ± 9.1	0	0	34.7 ± 4.7	0
Control ^b^	0	0	436.9 ± 3.8	10.2 ± 1.4	0	13.3 ± 9.4	0
Control ^c^	0	0	335.5 ± 3.8	42.4 ± 2.1	22.2 ± 3.2	16.2 ± 4.4	0
Control ^d^	100	100	0.9 ± 0.07	100	69.7 ± 1.6	100	100
Control ^e^	0	0	>300	22.1 ± 2.6	0	29.0 ± 3.8	0

^a^ Average of three replicates; ^b, c, d, e^ Control is fosthiazate, avermectin, fluopyram, and tioxazafen, respectively.

**Table 2 ijms-24-05773-t002:** The LC_50_ value of some target compounds against *A. besseyi and D. dipsaci*
^a^.

Compound	*A. besseyi*			*D. dipsaci*		
	LC_50_ μg/mL	Regression Equation	R	LC_50_ μg/mL	Regression Equation	R
**A1**	6.4 ± 0.6	Y = 3.9x + 1.8	0.95	3.6 ± 1.6	Y = 5.6x + 1.9	0.96
**A2**	5.0 ± 0.3	Y = 2.8x + 3.0	0.95	3.9 ± 0.4	Y = 3.7x + 2.8	0.99
**A3**	9.6 ± 4.5	Y = 2.1x + 3.0	0.95	4.6 ± 0.1	Y = 2.1x + 3.6	0.97
**A4**	11.3±6.3	Y = 4.5x + 0.3	0.96	3.5 ± 0.9	Y = 2.9x + 3.4	0.94
**A6**	3.8 ± 1.7	Y = 2.0x + 3.8	0.90	4.0 ± 0.4	Y = 4.0x + 2.6	0.93
**A7**	4.8 ± 0.4	Y = 2.5x + 3.3	0.90	2.7 ± 0.1	Y = 7.3x + 1.6	0.95
**A9**	13.6 ± 0.3	Y = 2.6x + 2.1	0.96	4.4 ± 0.2	Y = 3.6x + 2.7	0.97
**A10**	20.1 ± 1.6	Y = 4.1x − 0.3	0.93	3.6 ± 2.5	Y = 6.6x + 1.3	0.95
**A11**	15.3 ± 0.6	Y = 4.9x − 0.8	0.94	7.0 ± 1.7	Y = 2.8x + 2.6	0.90
**A13**	16.5 ± 2.4	Y = 2.5x + 1.9	0.94	7.2 ± 0.9	Y = 3.0x + 2.4	0.92
**A23**	12.3 ± 4.8	Y = 2.3x + 2.5	0.97	10.7 ± 0.6	Y = 2.3x + 2.6	0.96
**B1**	5.1 ± 2.1	Y = 4.1x + 2.1	0.91	4.5 ± 0.3	Y = 3.4x + 2.8	0.98
**B2**	6.7 ± 0.2	Y = 1.8x + 3.5	0.91	4.6 ± 0.6	Y = 3.6x + 2.6	0.96
**B3**	7.2 ± 2.5	Y = 2.9x + 2.5	0.96	3.2 ± 1.4	Y = 7.9x + 4.1	0.99
**B4**	6.4±0.7	Y = 2.7x + 2.8	0.92	3.7 ± 2.3	Y = 4.6x + 2.4	0.95
**B5**	5.1 ± 0.1	Y = 3.5x + 2.5	0.93	3.7 ± 2.1	Y = 6.7x + 1.2	0.94
**B6**	8.7 ± 2.7	Y = 3.8x + 1.4	0.97	4.2 ± 0.1	Y = 5.7x + 1.4	0.94
**B8**	6.2 ± 1.0	Y = 2.7x + 2.9	0.93	3.0 ± 0.2	Y = 4.5x + 2.9	0.91
**B9**	14.5 ± 1.9	Y = 1.2x + 3.6	0.93	4.2 ± 6.0	Y = 4.3x + 2.3	0.92
**B11**	14.2 ± 3.8	Y = 3.3x + 1.2	0.98	3.7 ± 0.1	Y = 7.2x + 0.9	0.98
Control ^b^	388.5 ± 6.0	Y = 4.8x − 7.4	0.94	333.3 ± 9.5	Y = 5.1x − 7.9	0.91
Control ^c^	56.8 ± 5.2	Y = 3.5x − 1.1	0.98	285.4 ± 5.4	Y = 4.9x − 7.0	0.90
Control ^d^	1.5 ± 0.1	Y = 2.5x + 4.5	0.96	0.8 ± 0.1	Y = 2.8x + 5.3	0.92
Control ^e^	142.9 ± 7.4	Y = 3.5x − 2.5	0.94	>300	-	-

^a^ Average of three replicates; ^b, c, d, e^ Control is fosthiazate, avermectin, fluopyram, and tioxazafen, respectively.

**Table 3 ijms-24-05773-t003:** Calculated binding free energies (kcal/mol) of **A1** and fosthiazate with AChE protein ^a^.

Compounds	ΔEvdw	ΔEele	ΔEMM	ΔGsol	ΔEbind	−TΔS	ΔGbind
**A1**	−141.47	−14.19	−155.66	111.69	−43.97	5.89	−38.08
**Fosthiazate**	−103.39	−43.35	−146.74	88.02	−58.72	31.25	−27.47

^a^: Free energies and their components were obtained from MM-PBSA calculations.

**Table 4 ijms-24-05773-t004:** List of primers used in this study.

Genes	Sequences (5′–3′)
*AChE 1-F*	ACCGGCACTTTCACCGAATA
*AChE 1-R*	AGCCCCAGGGAATATGGGAT
*AChE 2-F*	TGTCCGACATGCTTTGCTTA
*AChE 2-R*	ATCCCCAACAAATCGGGCAA
*ChAT-F*	TGCGACACTGTGTTCAATGC
*ChAT-R*	CGGAAGCAATAATGCAGGGC
*Actin-F*	GAAAGAGGGCCGGAAGAG
*Actin-R*	AGATCGTCCGCGACATAAAG

## Data Availability

All data generated in this study are presented in the current manuscript. No new datasets were generated. Data are available upon request from the corresponding author.

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
