# Peer review of "Discovery of 1,2,4-Oxadiazole Derivatives Containing Haloalkyl as Potential Acetylcholine Receptor Nematicides"

_ijms, 2023, doi:10.3390/ijms24065773_

Round 1

Reviewer 1 Report

The manuscript requires strict editing and most of the manuscript has to be re written. The introduction has to be re written as several language and scientific related flaws are clearly seen. The abstract, materials and methods and results have need editing with English language professionals to make the manuscript sound scientific. 

No doubt the work is interesting but the presentation of the work in the form of a manuscript is not up to the mark. I request strict editing of the manuscript to improve its scientific content.  

Author Response

Dear reviewer,

Thank you very much for your attention throughout the evaluation process of our manuscript “Discovery of 1,2,4-Oxadiazole Derivatives Containing Haloalkyl as Potential Acetylcholine Receptor Nematocides” submitted to International Journal of Molecular Sciences. The manuscript has been carefully modified following your comments. Each comment has been addressed and answered point by point in the revised manuscript. These revisions are colored red in the revised manuscript and also listed below for your reference.

The revised manuscript is hereby submitted to be considered for publication. Thank you for your kind co-operation and understanding throughout the evaluation process on our manuscript. We believe that the revised manuscript is more suitable for publication in International Journal of Molecular Sciences than before.

Response comments

  1. The manuscript requires strict editing and most of the manuscript has to be re written. The introduction has to be rewritten as several language and scientific related flaws are clearly seen. The abstract, materials and methods and results have need editing with English language professionals to make the manuscript sound scientific.

Answer: Thank you for your suggestions. According to your comment, the introduction has been written according to your suggestion, the contents was shown in revised manuscript. The abstract, materials and methods and results has been checked by a professional English editing service from MDPI (english-edited-61528) for perfection in English writing and grammars, these revisions are colored red in the revised manuscript and also listed in part of other corrections made by authors. We believe that the revised manuscript is more suitable for publication on International Journal of Molecular Sciences than before.

  1. No doubt the work is interesting but the presentation of the work in the form of a manuscript is not up to the mark. I request strict editing of the manuscript to improve its scientific content.

Answer: Thanks for your good comment on our manuscript, and your recommendation is appreciated. The manuscript has been checked by a professional English editing service from MDPI (english-edited-61528) for perfection in English writing and grammars. The contents were shown in revised manuscript.

Reviewer 2 Report

The research group of Gan and co-workers presented the synthesis of 1,2,4-oxadiazole derivatives bearing halogens, and also studied their nematicidal activity.

The results will attract the scientific community's attention since several compounds showed outstanding biological activity. Nonetheless, I suggest improving the manuscript, here are some points to be attended:

  1. Line 45 says "cyclobutr" must say cylcobutrifluram
  2. Figure 3. Please order the products according to the electronic nature of the substituents. For instance: Group I: Electro-donating; Group II: Electron-withdrawing, etc. Please improve the presentation.
  3. Tables 1 and 2 showed a bunch of data, and as seen is difficult to follow the results. I suggest ranking from good to no-activity, instead of just numbering it from A1 to B17. Maybe the authors can use a graph to show the results in a more aesthetic way. Again, improve the presentation.
  4. Section 2.6.1. In order to compare the results is necessary to show the results obtained for the control (+), for instance: tioxazafen.
  5. Section 3.1.2 Please add the quantities used in the experiment. How many grams/milligrams, milliliters, etc?
  6. Sup Inf. Please include the 13C spectra for all the compounds, also should be included the HRMS indicating the error in ppm. Without these data, the manuscript should not be accepted.

Author Response

Dear reviewer,

Thank you very much for your attention throughout the evaluation process of our manuscript “Discovery of 1,2,4-Oxadiazole Derivatives Containing Haloalkyl as Potential Acetylcholine Receptor Nematocides” submitted to International Journal of Molecular Sciences. The manuscript has been carefully modified following your comments. Each comment has been addressed and answered point by point in the revised manuscript. These revisions are colored red in the revised manuscript and also listed below for your reference.

The revised manuscript is hereby submitted to be considered for publication. Thank you for your kind co-operation and understanding throughout the evaluation process on our manuscript. We believe that the revised manuscript is more suitable for publication in International Journal of Molecular Sciences than before. The response of your comments were listed in attachment file.

Reviewer 3 Report

Article: “Discovery of 1,2,4-Oxadiazole Derivatives Containing Haloalkyl as Potential Acetylcholine Receptor Nematocides”

Dear Authors, I read your article, the topic itself is very interesting and from my point of view the knowledge conteined in it might be useful in future usage. Additional note: the text was written quite sloppy, with punctuation errors and uneven font. But I have few questions and comments. I will comment them along with position in text.

Line 7: “Correspondence: Correspondence: gxh200719@163.com (X.G.)”, delete one correspondence

Line 9: “difficultly controlling” change into controlling dificulties

Line 9: Tioxazafen – the chemical structure and proper chemical name should be added in here

Line 10: change “effect” into “effects”

Line 11 and whole text: change nematoidal into nematocidal

Line 13: “Most” no capital letter in here

Line 14: “Bursaphelenchus xylophilus (B. xylophilus), Aphelenchoides besseyi (A. besseyi) and Ditylenchus dipsaci (D. dipsaci)” the names in brackets are obvious along with biological nomenclature and should be deleted in here. It is obvious that when we mention the genus name for the first time full name should me given, next time only shorten version

Line 16 and whole text: the way of LC50 notation should be unifided in whole text

Line 18: “tioxazafen (>300 μg/mL, 142.9, and >300 μg/mL, respectively)” twice the same value >300 μg/mL was given

Lines 16-19: the way of value (value and unit or only value and the unit after last value) notation should be unified in this fragment

Line 26: shape into size

Line 28-29: the same as above

Line 28: delete dot after world worldwide

Line 36: once you give chemical name and once the trade name “such as methyl
bromide, 1,2-dibromo-3-chloropropane and aldicarb” please unify.

Line 66: in vitro need to be written with italics

Line 68: “A23” in brackets the chemical structure or chemical name need to be added, as the figure 3 which explains the abbreviation is located on next page

Line 76: “Forty-eight” no capital letter should be given here

Figure 3: I propose that A23 was somehow highlighted here as the whole manuscript in rounding around that compound

Line 86: As it was shown in Table 1

Line 93: “marvelous” change into significant

Line 101: unify as it was mentioned before

Table 1: add the letter “L” into control word, the statistics should be give in Table 1, not only values

Table 2: change “Compd” into “compound

Line 122: “compounds substituted” into “compounds with substituted”

Line 147: “a total of 58292” total of what? DEGs? Should be added here

Line 147: CK explain the abbreviation and give the specific information about used control, as before 4 controls were used

Line 164: delete capital letters from words: cholinergic, metabolism, drug, drug

Line 170: “kill nematodes by interfering nerve signals of B. xylophilus” I would rather said that it might act trough nervous system of B. xylophilus, further studies are needed in here. Because if the gene encoding cholinergic synapses are up-regulated maybe more acetylcholine is also secreted? By the way this results need to checked through e.g. realtime PCR technique to confirm it. Or maby less acetylcholine esterase? Because the graph in figure 6 does not show statistically significant changes. By the way, specific statistics with their sizes should be added.

Figure 6: as mentioned before

Line 254: change into “general procedure for the synthesis of compounds A1-A31 and B1-B17

Line 263: “B. xylophilus, D. dipsaci and A. besseyi were cultivated from potato dextrose agar-Botrytis cinerea” I think that not from but on?

Line 268: “pipette gun” you meant pipette?

Line 273: “same concentration” what concentration?

Line 280: Transcriptome Profiling – what was the control in here?

Line 282: “B. xylophilus was stored” whole body? Samples? RNA? DNA? Tissues?

Line 286: “High-quality mRNA was enriched” how you measures quality? What was the concentration of mRNA? From what company the reagents were obtained?

Line 297: what was the coverage?

Line 341: the conclusion are way to far in my opinion…

Discussion: in my opinion there is no discussion, only the summary of results was given here.

Author Response

Thank you very much for your attention throughout the evaluation process of our manuscript “Discovery of 1,2,4-Oxadiazole Derivatives Containing Haloalkyl as Potential Acetylcholine Receptor Nematocides” submitted to International Journal of Molecular Sciences. The manuscript has been carefully modified following your comments. Each comment has been addressed and answered point by point in the revised manuscript. These revisions are colored red in the revised manuscript and also listed below for your reference.

The revised manuscript is hereby submitted to be considered for publication. Thank you for your kind co-operation and understanding throughout the evaluation process on our manuscript. We believe that the revised manuscript is more suitable for publication in International Journal of Molecular Sciences than before. The response of your comments were listed in attachement file.

Round 2

Reviewer 3 Report

Thank you for preoaring the review. I have few comments.

Abstract: tioxazafen (>300, 142.9, and >300 μg/mL, respectively) – are you sure that the value “>300” should be stated twice?

Line 26: change the word shape into word size.

2.2. Nematoidal Activity - nematocidal

4List of primers used in this study- punctuation

Conclusions: “which could sever as cheap” did you mean serve?

Figure 8 – give the name of used statistics. Also add paragraph called ‘statistical analysis’ and please write there all of the statistical test that were used in analysis

In my opinion there is still lack of proper discussion in the text.

Author Response

Dear reviewer, Thank you very much for your attention throughout the evaluation process of our manuscript “Discovery of 1,2,4-Oxadiazole Derivatives Containing Haloalkyl as Potential Acetylcholine Receptor Nematicides” submitted to International Journal of Molecular Sciences. The manuscript has been carefully modified following your comments. Each comment has been addressed and answered point by point in the revised manuscript. These revisions are colored red in the revised manuscript and also listed below for your reference. The revised manuscript is hereby submitted to be considered for publication. Thank you for your kind co-operation and understanding throughout the evaluation process on our manuscript. We believe that the revised manuscript is more suitable for publication in International Journal of Molecular Sciences than before. The contents of the revised was shown in attachement.
